# Exploratory Analyses of Circulating Neoplastic-Immune Hybrid Cells as Prognostic Biomarkers in Advanced Intrahepatic Cholangiocarcinoma

**DOI:** 10.3390/ijms25179198

**Published:** 2024-08-24

**Authors:** Ranish K. Patel, Michael S. Parappilly, Brett S. Walker, Robert T. Heussner, Alice Fung, Young Hwan Chang, Adel Kardosh, Charles D. Lopez, Skye C. Mayo, Melissa H. Wong

**Affiliations:** 1Department of Surgery, Division of Surgical Oncology, Oregon Health & Science University (OHSU), Portland, OR 97239, USA; patera@ohsu.edu (R.K.P.);; 2Department of Cell, Developmental, and Cancer Biology, Oregon Health & Science University (OHSU), Portland, OR 97201, USA; 3Department of Biomedical Engineering, Oregon Health & Science University (OHSU), Portland, OR 97201, USA; 4Department of Diagnostic Radiology, Oregon Health & Science University (OHSU), Portland, OR 97239, USA; 5Knight Cancer Institute, Oregon Health & Science University (OHSU), Portland, OR 97201, USA; 6Department of Medicine, Division of Medical Oncology, Oregon Health & Science University (OHSU), Portland, OR 97239, USA

**Keywords:** intrahepatic cholangiocarcinoma, circulating hybrid cells, circulating tumor cells, cancer biomarker

## Abstract

Existing clinical biomarkers do not reliably predict treatment response or disease progression in patients with advanced intrahepatic cholangiocarcinoma (ICC). Circulating neoplastic-immune hybrid cells (CHCs) have great promise as a blood-based biomarker for patients with advanced ICC. Peripheral blood specimens were longitudinally collected from patients with advanced ICC enrolled in the HELIX-1 phase II clinical trial (NCT04251715). CHCs were identified by co-expression of pan-cytokeratin (CK) and CD45, and levels were correlated to patient clinical disease course. Unsupervised machine learning was then performed to extract their morphological features to compare them across disease courses. Five patients were included in this study, with a median of nine specimens collected per patient. A median of 13.5 CHCs per 50,000 peripheral blood mononuclear cells were identified at baseline, and levels decreased to zero following the initiation of treatment in all patients. Counts remained undetectable in three patients who demonstrated end-of-trial clinical treatment response and conversely increased in two patients with evidence of therapeutic resistance. In the post-trial surveillance period, interval counts increased prior to or at the time of clinical progression in three patients and remain undetectable in one patient with continued long-term disease stability. Using our machine learning platform, treatment-resistant CHCs exhibited upregulation of CK and downregulation of CD45 relative to treatment-responsive CHCs. CHCs represent a promising blood-based biomarker to supplement traditional radiographic and biochemical measures.

## 1. Introduction

Intrahepatic cholangiocarcinoma (ICC) represents the second most common primary liver cancer in the United States, accounting for 10–20% of all primary liver cancers [1]. Prognosis remains dismal, with an estimated median overall survival of ~12 months from diagnosis, even in an era of modern surgical and systemic therapies [2,3].

Given that the majority of patients diagnosed with ICC are not candidates for tumor resection [4], new multidisciplinary therapeutic strategies, including liver-directed therapies delivered through surgically placed hepatic arterial infusion (HAI) pumps, are being investigated that personalize and tailor treatments to individual patients and their unique tumor biology, as a means to improve survival [5,6,7,8,9,10]. However, improved biomarkers to measure disease burden are needed to rapidly assess treatment response and to guide clinical decision-making. Carbohydrate antigen 19-9 (CA 19-9) remains the most established blood-based analyte, though is limited by the lack of a validated cut-off, and its non-specificity to cancer (i.e., is frequently elevated by a variety of non-malignant pancreaticobiliary disorders) [11,12,13,14]. Additionally, it is estimated that 5–10% of the general population does not express the Lewis blood group antigen necessary to produce CA 19-9 [15]. Currently, cross-sectional contrast-enhanced computed tomography (CT) or magnetic resonance (MRI) imaging represent the standard-of-care radiographic modalities for the assessment of treatment response. These imaging techniques heavily rely on size measurements as an imperfect surrogate for treatment response [16,17]. 

Circulating neoplastic-immune hybrid cells (CHCs) are a novel cancer cell population found in the peripheral blood of patients across a myriad of cancer types, including cholangiocarcinoma [18,19,20,21,22,23,24,25,26,27]. These hybrid cells express both tumor and immune phenotypes (identified by co-expression of pan-cytokeratin [CK^+^] and CD45^+^) and are distinct from conventional circulating tumor cells (CTCs), which lack immune phenotypes (CK^+^/CD45) [20,21,22,27,28]. In in vitro models, neoplastic-immune hybrids can form through spontaneous cell fusion between bone marrow-derived and tumor cells, resulting in hybrids that can replicate, have enhanced motility, and are more tumorigenic when compared to unfused tumor cells [21,29]. A role for hybrid generation is further supported in murine models of cancer and human tumors with detection of neoplastic cells co-expressing tumor and immune proteins and pointing to the possibility of an important role within the metastatic cascade [20,21,22,25,27]. Emerging data indicate that CHC levels, as assessed by flow cytometry and fluorescent immunohistochemistry, have significant translational value as a neoplastic biomarker. In patients with pancreatic ductal adenocarcinoma, CHC levels correlate with disease stage and overall survival, and the cells themselves harbor oncogenic KRAS mutations seen within the primary tumor from which they derive [21]. In patients with rectal and esophageal adenocarcinoma, pre-operative CHC levels successfully discriminate between those with pathologic complete and incomplete responses to neoadjuvant therapy [27]. In patients with colorectal liver metastases, longitudinal CHC numbers correlate with response to multimodality treatment and appear to increase prior to evidence of disease progression on cross-sectional imaging [27]. Notably, these data are difficult to reproduce in CTCs, owing to their relative rarity; CHCs are frequently detected at levels an order of magnitude greater than CTCs [19,20,21,22,25,27,30,31].

The role of CHCs as translational biomarkers in gastrointestinal malignancies continues to develop, and their utility in ICC remains unexplored. In the present study, we prospectively collected longitudinal peripheral blood specimens from patients with advanced, unresectable ICC, and investigated the utility of CHCs as a biomarker to gauge therapeutic efficacy from a multimodality treatment paradigm which integrates surgical HAI therapy.

## 2. Results

### 2.1. Patient Demographics

Five patients met inclusion criteria and were included in the study group cohort. The median age was 60 years; three (60%) patients were female and two (40%) were male. All five patients self-reported white and non-Hispanic race and ethnicity, respectively. Four (80%) patients were treatment-naïve prior to the initiation of trial protocol-directed therapy (and thus baseline peripheral blood specimens were collected prior to exposure to any therapy), and a single patient received one cycle of gemcitabine/cisplatin prior to trial enrollment. None of the enrolled patients ultimately demonstrated sufficient down-staging of disease in order to qualify for surgical resection. Longitudinal, serial blood specimens were collected from all five patients. Notably, blood samples were not available for analysis for one patient in the post-trial period. Thus, the median follow-up time in the cohort was 22.7 months (range 6.0–27.5) from trial enrollment. A median of 7 (range 6–7) blood samples were collected per patient while receiving trial-directed therapy, while a median of 2 (range 0–3) blood samples were collected per patient in the post-trial period (Table 1). 

### 2.2. CHC Levels Provided Real-Time Insight into Treatment Response to Systemic and HAI Therapies

At baseline, all five patients had detectable levels of CHCs (CK^+^/CD45^+^) (Figure 1A and Appendix A) with a median count of 13.5 CHCs/50,000 peripheral blood mononuclear cells (PBMCs; range 7.6–15.5); no CTCs (CK^+^/CD45^−^) were identified in any patient. The median CA 19-9 level was 20.5 U/mL (range 6.3–830.3) (Table 1). Baseline CHC counts in the study cohort were significantly higher when compared to healthy subjects (*n* = 15; median 0 CHCs (range 0–1.67); *p* < 0.0001) (Figure 1B). Additionally, CHC counts remained zero or near-zero when followed longitudinally in healthy subjects (Figure 1C). In response to trial-treatment initiation, CHC levels decreased to undetectable from baseline levels in all five patients (Table 2, Figure 2). This correlated with decreasing tumor burden (as determined by cross-sectional imaging measurements, Appendix A) as well as down-trending CA 19-9 levels for three patients (Patients 1, 2, and 4; although Patient 2 first had an initial marked increase in CA 19-9 levels upon treatment initiation). In two patients (Patients 3 and 5), tumor burden was stable or mildly increased while CA 19-9 levels increased, as CHC levels dropped to zero.

CHC levels remained undetectable in three patients until the completion of trial-directed therapy (Patients 2, 3, and 4), all of whom demonstrated clinical evidence of therapeutic response with decreased or stable tumor burden on cross-sectional imaging and decreased CA 19-9 levels. Two patients (Patients 1 and 5) demonstrated therapeutic resistance by the end of trial, as evidenced by tumor stability or growth and/or an up-trending CA 19-9; in both cases, there were concordant increases in CHC counts which temporally matched the clinical evidence of resistance to ongoing therapy.

### 2.3. CHC Levels Increased with Long-Term Disease Progression and Resistance

Given that CHC trends correlated with response to treatment while receiving trial-directed therapy, we were interested in evaluating whether CHC levels could be utilized in longer-term surveillance to capture disease progression. Thus, peripheral blood samples were collected from four of five patients during the post-trial period (Table 2, Figure 3). Therapeutic regimens were personalized for each patient at the direction of their multidisciplinary care teams; four patients were continued on concurrent HAI-floxuridine and systemic mFOLFIRI and a single patient who was transitioned to concurrent HAI-floxuridine and systemic gemcitabine and oxaliplatin; further treatment modifications for each patient during the post-trial period are detailed in Figure 3 and Appendix A. 

Three patients (Patients 2, 4, and 5) experienced long-term tumor response and/or stability, though ultimately demonstrated disease progression as indicated by up-trending CA 19-9 levels, increased tumor burden on cross-sectional imaging, and/or the development of new FDG-avid hepatic lesions on positron emission tomography (PET) imaging. In all three patients, there were interval increases in CHC counts that temporally matched the first evidence of disease recrudescence; interval counts increased by a median of 7.6 cells (range 2.0–49.6). Notably, Patient 2 experienced an elevation in CHC levels that preceded clinical indicators of disease regrowth on imaging or CA 19-9 levels, then subsequently decreased to zero after HAI treatment was re-initiated. Additionally, Patients 4 and 5 both developed biliary sclerosis, which were thought to be possible complications related to floxuridine-based HAI therapy. In both instances, CA 19-9 levels significantly increased despite no radiographic evidence of tumor growth and subsequently down-trended after initiating treatment with steroids and cessation of HAI therapy.

Patient 3 remained the lone patient without any clinical evidence of tumor progression during the post-trial surveillance period; their cross-sectional imaging tumor burden continued to decrease and ultimately stabilized with steadily down-trending CA 19-9 levels. Concordantly, CHCs remained undetectable throughout the post-trial period. 

### 2.4. Revealing CHC Morphology Changes through Representation Learning in Treatment Response

Given that CHCs consistently correlate with disease status, we further evaluated whether their morphology changed during periods of response and resistance to therapy. We categorized all identified CHCs from our study cohort into two classes: *treatment-responsive CHCs*, defined as cells detected from onset of treatment until the time when CHCs first became undetectable, and *treatment-resistant CHCs*, defined as recurrent cells identified following periods of undetectable CHC counts. Employing a β-variational autoencoder (VAE) model that our group has previously validated [32], we extracted latent features from CHC images capturing their cellular morphologies, marker intensity levels, and staining characteristics (Figure 4 inset). The UMAP visualization of the latent embeddings demonstrates a moderate overlap across patient classes (Figure 4A), with distinct clustering of treatment-responsive and treatment-resistant CHCs (Figure 4B). Specifically, treatment-responsive CHCs exhibited elevated CD45 expression and decreased CK expression, while treatment-resistant CHCs showed the opposite pattern (i.e., decreased CD45 and elevated CK expression) (Figure 4C). Notably, the CHCs from the lone patient (Patient 3) without clinical evidence of tumor progression demonstrate phenotypic similarities to treatment-responsive CHCs observed in the remaining four patients, highlighting potential clinical implications of CHC phenotyping in treatment monitoring.

## 3. Discussion

While tumor resection remains the only curative-intent option for patients with ICC, the vast majority of patients present with unresectable, advanced disease; thus, the development of effective systemic and surgical liver-directed therapies, which can control disease burden while preserving liver function, is of paramount importance [4,33,34]. However, as these therapies continue to evolve, there is an additional urgent need for sensitive biomarkers reflective of response so that treatment can be both optimized and personalized to individual patients and their tumor biology. The data from this study indicate that CHCs can provide invaluable, real-time assessments of treatment efficacy, can serve as a surveillance marker, and may phenotypically evolve in response to multimodal therapies. 

While receiving trial-directed therapy, CHC levels were reflective of response trends, even when incongruent or equivocal data arose from CA 19-9 or radiographic measures. This was well demonstrated in the case of Patient 3, who saw CHC counts drop to undetectable levels, while concurrent cross-sectional imaging did not definitively reflect this by size measurements, and CA 19-9 levels initially spiked before down-trending. However, given that this patient’s disease ultimately demonstrated continued response and stability at long-term follow-up, it appears that the early and rapid decrease in CHCs more accurately reflected disease response when compared to the standard-of-care measures. This highlights an all-too-common clinical scenario, where real-time monitoring of CHC levels may provide better insight on tumor activity. As an inherently tumor-derived cell population, CHCs may more directly reflect intrinsic tumor biological response behaviors, while also being more resistant to the natural variance seen with current biomarkers [11,12,13,14,16]. CHC analyses may therefore be best leveraged as supplements to traditional response measures to provide a more comprehensive functional assessment of treatment response in patients with advanced ICC. 

Importantly, long-term surveillance and monitoring of disease relapse in those with unresectable, advanced ICC can be challenging, given limitations with CA 19-9 reliability (particularly well demonstrated by increased levels in our cohort relating to HAI-related biliary sclerosis) as well as the costs and practicality of repeated cross-sectional imaging. While there were sharp increases in CHC counts at the time of disease progression in three patients, a rise in CHC counts preceding evidence of a new focus of FDG-avidity on PET imaging (despite a stable tumor size) in Patient 2 is notable. This encapsulates the inherent limitations of standard radiographic size measurements and reflects the potential for CHCs to be utilized as more sensitive, functional surveillance biomarkers which perhaps relate more directly to tumor viability and activity, as compared to size measurements. This will become particularly relevant as immune-modulating therapies (e.g., durvalumab and pembrolizumab) are integrated into systemic regimens for ICC, as these treatments can paradoxically increase tumor size with treatment effect [2,35]. However, the utility of CHCs as biomarkers in the setting of immunotherapies, which may confound CHC generation and dissemination, remains unexplored. Together, these data highlight the potential for CHCs to signal treatment resistance earlier and more reliably than conventional techniques, which may identify opportunities to change and optimize treatment modalities to improve patient survival.

Furthermore, we demonstrate that employing β-VAE for representation learning can unveil subtle morphological variations in disseminated tumor cells that are undetectable to the eye. Our analysis reveals distinct phenotypic differences between treatment-responsive and resistant CHCs, suggesting a potential reflection of evolving tumor biology under treatment. Notably, the treatment-resistant CHC subset exhibits an enrichment of cells with upregulated CK expression and downregulated CD45 expression across the patient cohort, perhaps representing a distinct cellular identity with more tumor-like characteristics and diminished immune features. Utilizing unsupervised machine learning, we can extract image features that go beyond conventional metrics like mean intensity or eccentricity, enabling the discovery and comparison of visually distinct phenotypes. For example, the β-VAE identified a punctate CK staining pattern in treatment-resistant CHCs from Patients 2 and 5 (Figure 4 inset). While larger sample sizes and comprehensive phenotypic and genotypic profiling are necessary for validation, our quantitative image analyses highlight the potential of characterizing CHCs to gain meaningful qualitative insights into treatment response versus resistance mechanism.

CHCs were found in all five of our patients at baseline while CTCs were not identified at baseline in any of the study patients. The lack of CTC detection in our study is consistent with previous work in ICC, which only detected CTCs in a fraction of patients and, when present, at exceedingly low numbers; a 7.5 mL blood sample might yield as low as one detectable CTC [30,36,37]. Conversely, in line with previous work [20,21,25,27,38], CHCs were robustly and reliably detected in our patient cohort with a median of ~14 CHCs per 50,000 sampled PBMCs, which reflects only a small fraction of blood volume. However, current detection methods for CHCs are limited compared to those available for CTCs. While multiple CTC detection platforms exist [39,40,41,42,43,44], the current gold standard is the FDA-approved CellSearch system [45], which specifically excludes CD45^+^ cells to facilitate high-throughput identification of CTCs. No such detection platform exists for CHCs and high-throughput, automated enumeration is not yet possible. The development of an automated platform for CHC detection and quantification remains an active area of investigation. Other liquid biomarkers, such as circulating tumor DNA (ctDNA) have also demonstrated promise as in ICC, although primarily to identify mutational aberrations, or in the perioperative setting to predict disease clearance and track recurrence risk [46,47,48]. Data evaluating the utility of ctDNA as real-time markers of therapeutic response for those with locally advanced, unresectable disease are limited [49,50]. ctDNA is rare in circulation and has a short half-life, and automated detection platforms have different limits of detection, making negative results challenging to interpret, particularly in the context of assessing treatment response [51]. In this context, CHCs may theoretically be better suited as a more-sensitive biomarker that is reliably detectable in early-stage disease, though there are no studies comparing CHCs and ctDNA detection at the present. 

There are important limitations to this study. First, there were a small number of patients included as this was reflective of the relative rarity of this malignancy and the number of patients who were eligible and successfully enrolled in the trial. However, multiple blood samples were collected per patient, to build redundancy in these data and to improve the validity of the observed trends. Secondly, while many blood samples were collected at similar time-points across the study cohort, the timing of sample collection was not standardized and was influenced by patient availability. This was most notable during the extended periods of disease stability for Patients 2, 4, and 5, where we were unable to evaluate interval CHC counts at earlier time-points; however, this was completed for Patient 3. This also limited our ability to make direct comparisons to CA 19-9 levels, which were frequently collected at short intervals as a part of standard clinical care, though these levels trended inconsistently with disease biology. A more rigorous comparative analysis between CHCs and CA 19-9 remains an important future step in this work. Finally, given a limited sample size, these results are descriptive in nature. This represents an exploratory study not powered to facilitate more-robust statistical analyses to identify differences between patients or specific CHC value cut-offs or thresholds. As the data supporting CHCs as translational cancer biomarkers continue to grow, it would be prudent to ultimately translate the findings of these analyses into a more formal clinical assay; for example, defining specific CHC thresholds and evaluating whether they are universal or specific to different malignancies remains unexplored. Larger prospective data are needed to further rigorously validate the findings observed in this study.

## 4. Materials and Methods

### 4.1. Human Patient Specimens

All human peripheral blood specimens were collected between April 2021 and December 2023 from patients who successfully enrolled and participated in the HELIX-1 clinical trial (NCT04251715), which is an open-label, single-institution phase II trial of induction systemic mFOLFIRINOX (folinic acid, 5-fluorouracil, irinotecan, and oxaliplatin), followed by surgical staging and placement of an HAI pump and subsequent administration of concurrent HAI-floxuridine and systemic mFOLFIRI (folinic acid, 5-fluorouracil, and irinotecan), for patients with liver-confined unresectable ICC. Specimens were collected under approved protocols in accordance with the ethical requirements and regulations of the Oregon Health & Science University (OHSU) institutional review board and with informed consent from all participants.

### 4.2. Patient Cohort and Schedule of Therapy

All included patients met the following criteria: unresectable liver-only ICC, no evidence of extrahepatic metastatic disease (as determined by imaging and surgical staging laparoscopy at initial screening), no evidence of leukopenia, microsatellite-stable/mismatch repair-proficient tumor biology, and no prior treatment with mFOLFOX (folinic acid, 5-fluorouracil, and oxaliplatin) or mFOLFIRINOX, liver-directed external beam radiation therapy, or chemoradiation therapy. After trial protocol-directed treatment concluded, all participants underwent clinical restaging and were transitioned to standard-of-care therapies, as directed by their multidisciplinary oncology team. Post-trial-treatment was personalized to each participant and was not standardized across the study cohort. To protect patient identity, the order of patients presented is not reflective of the order that patients enrolled in the trial. Additionally, peripheral blood samples were collected from healthy subjects to be utilized as controls, and follow-up samples were collected in a limited number of healthy subjects to be utilized as longitudinal controls. Healthy subjects were recruited to this study either as self-reported healthy volunteers or as self-reported healthy volunteers who underwent negative standard screening colonoscopies. Blood samples were collected immediately prior to proceeding with colonoscopy.

### 4.3. Specimen Collection and Processing 

Serial peripheral blood specimens were longitudinally and prospectively collected from each patient during and, as available, in the follow-up period after clinical trial protocol completion. All blood specimens were collected as part of scheduled standard-of-care blood draws, immediately prior to initiation of a scheduled visit treatment. Patient peripheral blood (10–20 mL) was collected into heparinized vacutainer tubes (BD Biosciences, Franklin Lakes, NJ, USA). PBMCs were isolated using standard density centrifugation with Ficoll-Paque (GE Healthcare, Chicago, IL, USA). PBMCs were adhered to poly-D-lysine-coated glass slides (Fisher Scientific, Hampton, NH, USA), incubated at 37 °C for 15 min, permeabilized with Triton-X, and fixed with 4% paraformaldehyde. 

### 4.4. Immunohistochemistry and Quantitative Imaging

PBMCs were then incubated in blocking buffer (2.5 M CaCl_2_, 1% Triton-X-100, 1% bovine serum albumin in phosphate-buffered saline), stained with fluorescent-conjugated antibodies against pan-cytokeratin (CK, AE1/AE3, Invitrogen, Waltham, MA, USA) and CD45 (HI30, Life Technologies, Carlsbad, CA, USA) and counterstained with 4,6-diamidino-2-phenylindole (DAPI). Single-antibody staining and background controls were evaluated in a region on the same slide. 

The stained PBMCs were digitally imaged using a Zeiss AxioScan. Z1 light microscope (Zeiss, Oberkochen, Germany). CHCs were identified by co-expression of CK and CD45 (CK^+^/CD45^+^); CTCs were identified by CK expression only (CK^+^/CD45^−^). Fluorescence histogram thresholds were established using the unstained cells as reference points. CHCs and CTCs were enumerated using a semi-automated approach, blinded to the clinical status of the specimen, using the Zeiss Efficient Navigation (ZEN) blue software (Zeiss, Germany, https://www.zeiss.com/microscopy/en/products/software/zeiss-zen-starter.html, accessed on 28 July 2024). For each peripheral blood sample, CHC numbers were reported normalized to 50,000 total PBMCs.

### 4.5. CHC Phenotyping with Representation Learning 

The PBMC images were segmented with Mesmer using DAPI as the nuclear marker and the projection of CK/CD45 as the membrane marker [52]. CHCs from the five patients were cropped from annotated images, padded to a size of (64, 64, 3) pixels, and normalized based on median expression levels of CK, CD45, and DAPI within each PBMC sample. Next, as we have previously validated, these CHC images were used to train a β-variational autoencoder (β-VAE) for extracting latent features [32,53], utilizing a convolutional encoder with five layers of 32, 64, 128, 256, and 512 filters, with a latent dimension of 64 and mirrored decoder. We employed a β scheduler during training, starting at β = 0.001 and increasing in intervals of 0.001 every batch until β = 2 was reached and remained until training was complete [54]. The model, built on the PyTorch Lightning platform, was trained on a single NVIDIA A40 GPU with a batch size of 16, learning rate of 0.0005, early stopping, and a maximum of 100 epochs. The subsequent CHC embeddings were visualized with a uniform manifold approximation and projection (UMAP) tool [55]. 

### 4.6. Statistical Analysis

Descriptive statistics for the clinicopathologic characteristics of the study patient cohort were compiled, with categorical variables reported as percentages and continuous variables as medians with ranges. A Mann–Whitney U test was utilized to compare CHC counts between enrolled patients at baseline and healthy control subjects using GraphPad Prism version 10.2.0. CHC levels were organized chronologically per patient and then ultimately compared to each patient’s clinical status, course, and pertinent laboratory values. Laboratory data, clinical history, treatment schedule, and multidisciplinary assessment of treatment response and resistance were extracted directly from the electronic health record. Radiographic tumor burden measurements were calculated by a board-certified radiologist through the sum of the diameters of measurable lesions per RECIST criteria (version 1.1) [56] and reported with accompanying narrative interpretation. 

## 5. Conclusions

Circulating hybrid cells are readily detectable in patients with advanced ICC and can be utilized as a blood-based biomarker to assess treatment response, resistance, and progression, to complement traditional biochemical and radiographic measures. This pilot study is the first to characterize CHCs in this patient population and serves as a first step towards validating CHCs as response biomarkers which can be utilized to tailor and optimize multimodal therapies to individual patients, in order to improve survival. 

## Figures and Tables

**Figure 1 ijms-25-09198-f001:**
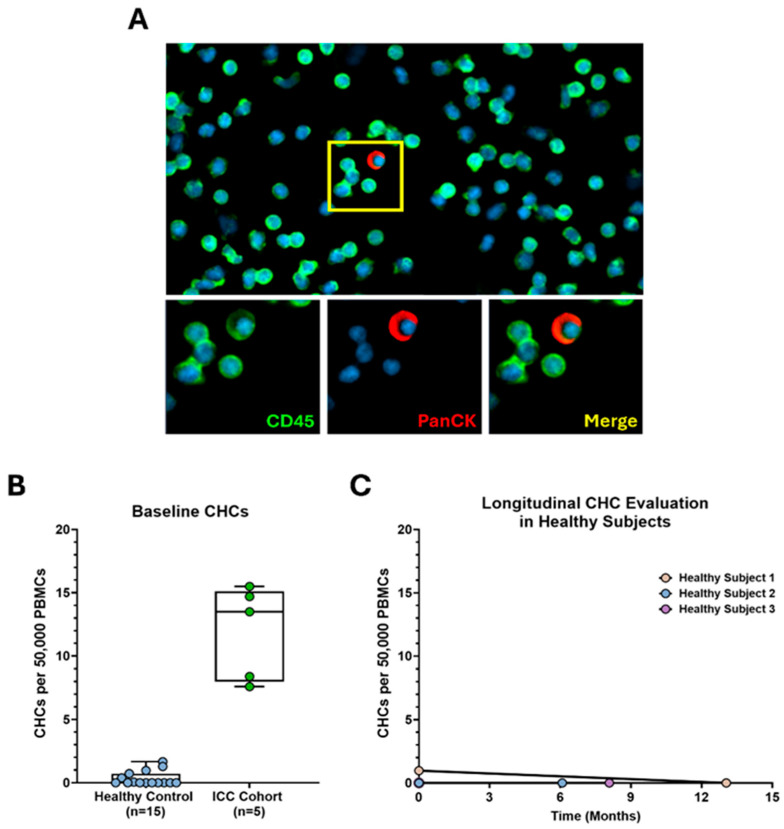
Circulating hybrid cells (CHCs) are detected in the peripheral blood of patients with advanced, unresectable intrahepatic cholangiocarcinoma (ICC), at higher levels than healthy subjects. (**A**) CHCs are identified via co-expression of CD45 (green) and pan-cytokeratin (CK; red). Nuclear DAPI staining is blue. (**B**) Baseline CHC counts of the study cohort are significantly higher compared to healthy controls (*p* < 0.0001). (**C**) CHC counts remain virtually undetectable in healthy subjects over time. PBMCs; peripheral blood mononuclear cells.

**Figure 2 ijms-25-09198-f002:**
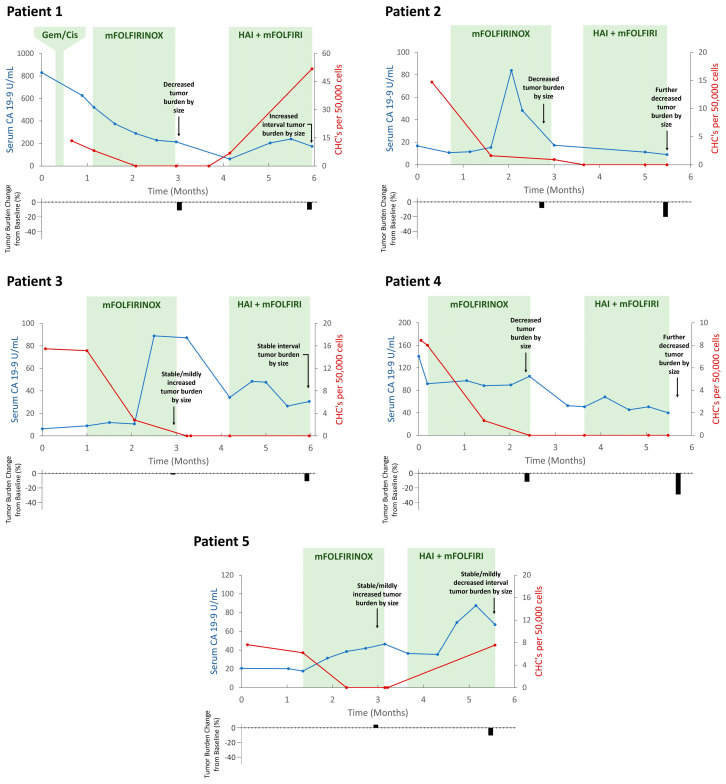
Circulating hybrid cell (CHC) levels correlate with therapeutic response and resistance while on trial protocol−directed therapy. Longitudinal evaluation of all five patients demonstrates an initial decrease in CHC counts after treatment initiation; this response was sustained in Patients 2, 3, and 4 who ultimately demonstrated sustained evidence of treatment response by the end of trial. Patients 1 and 5 demonstrated evidence of treatment resistance by the end of trial, which temporally correlated with increasing CHC levels. HAI, hepatic arterial infusion; mFOLFIRINOX, folinic acid, 5-fluorouracil, irinotecan, and oxaliplatin; mFOLFIRI, folinic acid, 5-flurouracil, irinotecan.

**Figure 3 ijms-25-09198-f003:**
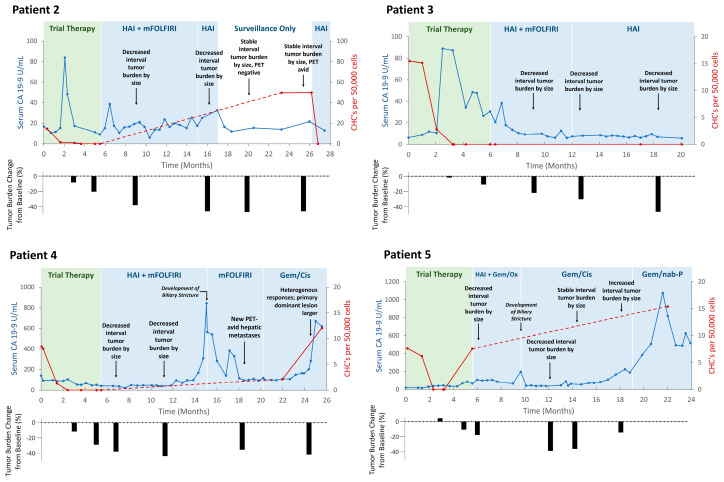
Circulating hybrid cell (CHC) levels correlate with long-term disease progression in the post−trial period. In Patients 2, 4, and 5, interval CHC counts increased near, or prior to, first evidence of disease progression following extended periods of disease response and/or stability. In Patient 3, who did not demonstrate evidence of disease progression, long-term CHC counts remained undetectably low. HAI, hepatic arterial infusion; mFOLFIRI, folinic acid, 5-flurouracil, irinotecan Gem/Ox, gemcitabine, and oxaliplatin; Gem/Cis, gemcitabine and cisplatin; Gem/nab-p, gemcitabine and nanoparticle albumin-bound paclitaxel.

**Figure 4 ijms-25-09198-f004:**
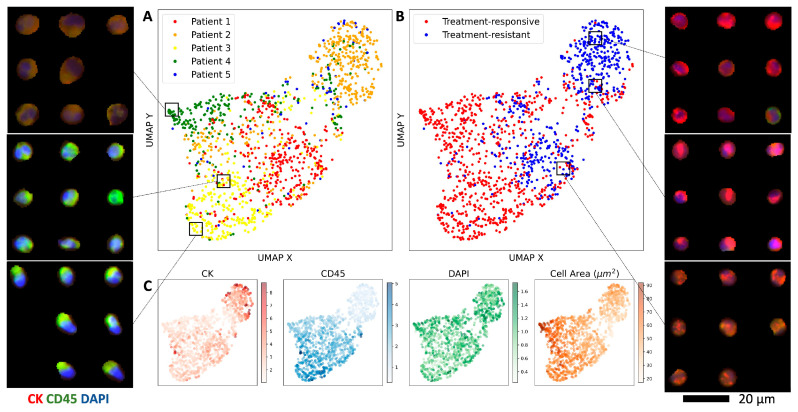
Treatment-resistant CHCs are phenotypically distinct from treatment-responsive CHCs. (**A**) UMAP of the β-VAE CHC embeddings colored by patient. (**B**) The same UMAP of the β-VAE CHC embeddings stratified by treatment response or resistance. (**C**) Relevant feature distributions across the UMAP, including normalized CK/CD45/DAPI expression and cell area. (**Inset**) CHC images plotted in the UMAP space reveal clusters with distinct morphological and staining characteristics.

**Table 1 ijms-25-09198-t001:** Cohort demographic and clinical characteristics.

Patients, no.	5
Median age, years (range)	60 (42–69)
BMI	30 (22.6–36.2)
Sex, no. (%)	
Female	3 (60)
Male	2 (40)
Race, ethnicity	
White, non-Hispanic, no. (%)	5 (100)
Median baseline CA 19-9, U/mL (range)	20.5 (6.3–830.3)
Median baseline CHC count, per 50,000 PBMCs (range)	13.5 (7.6–15.5)
Systemic therapy prior to trial enrollment	
Yes, no. (%)	1 (20)
Regimen, cycles	Gem/Cis, 1
Median dominant lesion size, cm (range)	9.8 (8.4–14.5)
Median number of intrahepatic lesions, no. (range)	9 (1–15)
Large vessel involvement, no. (%)	5 (100)
Median follow-up from trial enrollment, months (range)	22.7 (6.0–27.5)
Total number of blood samples collected, no.	42
Median number of blood samples collected per patient, no. (range)	9 (7–10)
Median number of blood samples collected per patient while on trial protocol treatment, no. (range)	7 (6–7)
Median number of blood samples collected per patient after completion of trial protocol treatment, no. (range)	2 (0–3)

Abbreviations: BMI, body mass index; CA 19-9, carbohydrate antigen 19-9; CHC, circulating hybrid cell; no., number; Gem/Cis, gemcitabine and cisplatin; PBMC, peripheral blood mononuclear cells.

**Table 2 ijms-25-09198-t002:** CHC levels in response to treatment and during post-trial surveillance.

	Trial-Directed Therapy		Post-Trial Surveillance	
	CHC Level Nadir after Treatment Initiation	CHC Levels by End of Treatment	Evidence of Progression	CHC Levels
Patient 2	Undetectable	Undetectable	Yes	Increased
Patient 3	Undetectable	Undetectable	No	Undetectable
Patient 4	Undetectable	Undetectable	Yes	Increased
Patient 1	Undetectable	Increased	N/A	N/A
Patient 5	Undetectable	Increased	Yes	Increased

Abbreviations: CHC, circulating hybrid cells. Response and resistance determined by multidisciplinary interpretation of standard-of-care carbohydrate antigen (CA) 19-9 and cross-sectional imaging measures (Appendix A) for each patient.

## Data Availability

The raw data supporting the conclusions of this article are available from the authors on request.

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
