# Peer review of "Exploratory Analyses of Circulating Neoplastic-Immune Hybrid Cells as Prognostic Biomarkers in Advanced Intrahepatic Cholangiocarcinoma"

_ijms, 2024, doi:10.3390/ijms25179198_

Round 1
Reviewer 1 Report
Comments and Suggestions for Authors
The authors of the manuscript analyzed the prognostic function of neoplastic-immune hybrid cells (CHC) within cholangiocarcinoma.
The work is very interesting, and the research group in question is a pioneer in the study of CHC.
However, some considerations are necessary:
- It is necessary to have more information on the healthy subjects cited in the "Results" and "Methods" sections and FIGURE 1c: are they healthy volunteers? From which cohort were they chosen? Are they health workers? Are they healthy relatives of the patients in the study?
- In all patients, except patient 3, the trend of CHC concentrations is theoretical due to the small number of serial samples. We cannot know if there were variations strictly in line or not with the modifications of the CA19-9 concentrations
- The patients analyzed in this work are small in number and often fragmented data. The Authors define this work as an "exploratory study". However, they should declare this nature of the study starting from the title or increase the cohort of patients that was already 5 in the works of the same group in previous years.
Reviewer 2 Report
Comments and Suggestions for Authors
Authors showed the results of analyses for circulating neoplastic-immune hybrid cells (CHCs) presenting both panCK and CD45 in advanced ICC patients. These results suggested that CHCs might be a blood-based biomarker to predict tumor progression. This study was interesting, but limited number of patients were included. Therefore, the claim was unclear. In Table 1, patients characteristics was shown. Authors should each 5 cases not but summarized form. Tumor size, tumor number and vessel invasion should be also shown. Representative images of ICC in each case are helpful for understanding the cases.
In appendix data, CHC1 to 6 were shown. In present study, only 5 cases were included. Which was correct?
Round 2
Reviewer 2 Report
Comments and Suggestions for Authors
Revised manuscript was well-addressed to the reviewers' comments was well-written.